# Echocardiographic Phase Detection Using Neural Networks

**Elisabeth Lane**[1]                                    Elisabeth.Lane@uwl.ac.uk

**Neda Azarmehr**[2]                                      nazarmehr@lincoln.ac.uk

**Jevgeni Jevsikov**[1]                                   jevgeni.jevsikov@uwl.ac.uk

**James P Howard**[2]                                              james@jph.am

**Matthew J Shun-shin**[2]                                    m@shun-shin.com

**Darrel P Francis**[2]                                     darrel@drfrancis.org

**Massoud Zolgharni**[1,2]                               massoud@zolgharni.com

[1] *School of Computing and Engineering, University of West London, London, United Kingdom*

[2] *National Heart and Lung Institute, Imperial College, London, United Kingdom*

## Abstract

Accurate identification of end-diastolic (ED) and end-systolic (ES) frames in echocardio-graphic cine loops is essential when measuring cardiac function. Manual selection by human experts is challenging and error prone. We present a deep neural network trained and tested on multi-centre patient data for accurate phase detection in apical four-chamber videos of arbitrary length, spanning several heartbeats, with performance indistinguishable from that of human experts.

**Keywords:** Echocardiography, Cardiac imaging, Deep learning, Phase detection.

## 1. Introduction

Mada et al. (2015) show that a 2-3 frame detection error in ES can elicit an approximate 10% difference in left ventricular (LV) function measurements; such as ejection fraction (EF) and global longitudinal strain (GLS). Subtle frame-on-frame spatial differences and complex temporal relationships present a barrier to consistent diagnosis. Therefore, automated methods could improve the consistency of echocardiographic quantification.

Eliminating the need for an accompanying ECG signal, Dezaki et al. (2017, 2019) and Fiorito et al. (2018) apply deep learning to automate phase detection for videos of one heartbeat; requiring pre-processing of the input. We target videos of arbitrary length and multiple beats because, in clinical practice, accuracy can be improved by averaging measurements over several consecutive cardiac cycles from the same acquisition.

## 2. Method

Three multi-centre apical 4-chamber (A4C) datasets were used in this study:

**PACS-dataset (training/testing):** Made **public** by the authors, from NHS Trust PACS Archives - Imperial College Healthcare. 1,000 videos containing 1-3 heartbeats. 2 expert annotations. Frame rate (fps): 23-102.

**Multibeat-dataset (testing): Private**, from St Mary's Hospital, London. 40 videos containing 10 heartbeats. 6 annotations by 5 experts. Frame rate (fps): 52-80.

**EchoNet-dataset (testing): Public**, Stanford University Hospital. 10,030 videos each

containing 1 heartbeat. Annotated by 1 expert. Frame rate (fps): 50.

The model architecture comprises a CNN unit (ResNet50 with ImageNet weights) for encoding spatial information, a RNN (2x LSTM) unit for decoding temporal information, from which return sequence is set to true then flattened and regressed through a Dense layer in chunks of 30 frames. A fixed-stride of 5 frames sliding window allows multiple predictions to be averaged for each input frame. A peak finding algorithm identifies discrete predictions for ED and ES relative to a predetermined threshold. Assigning target values of 1 to ED and 0 to ES time-points using the ground-truth labelled by an expert and a linear interpolation function, target outputs for all frames between the two events were defined.

Average absolute frame difference (aaFD) notation, where N is the number of events within the test dataset, measures the discrepancy between a labelled target $y^t$, (i.e. ED or ES), and a model prediction, $\hat{y}^t$:

$$aaFD = \frac{1}{N} \sum_{t=1}^{N} |y^t - \hat{y}^t|,$$

## 3. Results and Discussion

Table 1 details detection errors for all test videos in the PACS-dataset. The results indicate the discrepancy between Operator-1, i.e. the ground-truth, compared with model predictions and expert Operator-2.

Table 1: **PACS-dataset** results

| Model/operator | ED | | ES | | Detection time(s) |
|---|---|---|---|---|---|
| | aaFD | $\mu \pm \sigma$ | aaFD | $\mu \pm \sigma$ | |
| **Model** | 0.66 | -0.09±1.10 | 0.81 | 0.11±1.29 | 0.776±0.33 |
| **Operator-2 (inter-observer)** | 1.55 | -1.35±1.31 | 1.44 | -0.90±1.80 | 26±11 |

Table 2 illustrates results for the Multibeat-dataset. The mean difference between two annotations by the same expert (intra-observer variability) was -0.22±2.76 and 0.25±3.75. The range of mean differences between two different operators (inter-observer variability) was [-0.87, -5.51]±[2.29, 4.26] and [-0.97, -3.46]±[3.67, 4.68] for ED and ES events.

Table 3 details the results for the publicly available EchoNet-dataset when running predictions on the entire (10,000 videos), previously unseen, dataset.

Table 2: **Multibeat-dataset** results

| Model/operator | ED | | ES | |
|---|---|---|---|---|
| | aaFD | $\mu \pm \sigma$ | aaFD | $\mu \pm \sigma$ |
| **Operator-1a vs Operator-1b** | 1.96 | -0.22±2.76 | 1.90 | 0.25±3.75 |
| **Operator-1a vs Operator-2** | 2.65 | -1.22±4.26 | 3.67 | -2.25±4.68 |
| **Operator-1a vs Operator-3** | 5.82 | -5.51±3.77 | 4.80 | -4.46±3.77 |
| **Operator-1a vs Operator-4** | 1.72 | -0.87±2.29 | 2.01 | -0.97±3.48 |
| **Operator-1a vs Operator-5** | 3.27 | -2.96±2.57 | 4.11 | -3.64±3.67 |
| **Operator-1a vs model** | 2.62 | -1.34±3.27 | 1.86 | -0.31±3.37 |

Table 3: **EchoNet-dataset** results

| Model/operator | ED | | ES | |
|---|---|---|---|---|
| | aaFD | $\mu \pm \sigma$ | aaFD | $\mu \pm \sigma$ |
| **Model** | 2.30 | 0.16±3.56 | 3.49 | 2.64±3.59 |

Our model performance was assessed against three multi-centre datasets: one of which made public by the authors for the benefit of researchers and benchmarking of future studies (`intsav.github.io/phase_detection.html`), another already publicly available. We demonstrate the performance of the proposed model is akin to human experts; detection error is within the range of calculated inter-observer variability.

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
