# OpenReview forum: "Echocardiographic Phase Detection Using Neural Networks"
_MIDL.io/2021/Conference/Short — MIDL 2021 Poster_

### Official Review · Reviewer_hV1T · 2021-04-22

**Confidence:** 4
**Final Rating:** 3

**Summary:**

The paper describes a model for automatically identifying end-systole and end-diastole frames in echocardiography videos containing an arbitrary number of heart beats. The model is based on CNN/RNN modules followed by a regression to predict cardiac phase. The model is applied in a sliding (temporal) window fashion and the results of each phase prediction are averaged to give the final result.

**Strengths:**

The model is evaluated on multi-centre datasets, specifically echocardiography data from three separate sources. This type of evaluation on external datasets is important for clinical translation.
The framework proposed is able to handle video sequences of arbitrary length.
The authors commit to publicly sharing one of the datasets used.


**Weaknesses:**

The motivation for being able to process videos of arbitrary length is not clearly stated.
The means by which ED/ES frames are identified from the cardiac phase curves predicted by the model is not made clear.


**Deanonymize Review:**

no

**Detailed Comments:**

Please define the abbreviations ED and ES the first time they are used.
In Section 1 first sentence the authors mention that errors in identifying ED/ES frames can cause errors in “cardiac measurements”. Can they be more specific about which measurements? Presumably ejection fraction?
As noted above, please briefly mention why being able to process videos of arbitrary length is clinically useful. Are they common in clinical practise?
As noted above, the means by which the final ED/ES frame prediction is made is not clear. The use of a regression model implies that the output of the CNN/RNN is a sequence of continuous values representing cardiac phase. How is this converted into discrete predictions of ED/ES frames?
In the equation in Section 2, presumably N is the number of ED/ES frames in the sequence? If so, please state this.


**Justification Of The Rating:**

I like the use of external validation sets and (as far as I know) the proposal of a model that can handle videos of arbitrary length is new. If the authors could address the issues of clarity highlighted above I think the paper can be a useful contribution.

**Paper Type:**

both

**Special Issue:**

no

---

### Official Review · Reviewer_Pvbi · 2021-04-28

**Confidence:** 4
**Final Rating:** 3

**Summary:**

The authors propose a novel model to estimate the cardiac phase from echocardiographic videos. The model predicts the ED and the ES frames, with an average inter frame difference < 2 frames in average. The paper is well written and the methods are presented clearly; the results could be described more clearly though.



**Strengths:**

+ A novel architecture that seems to work well in a variety of datasets
+ Multiple datasets used, with a combination of public and private data coming from different centers and manufacturers
+ Accuracy comparable to human performance in a fraction of the time

**Weaknesses:**

* Authors claim that results are indistinguishable from humans but there is no statistical significance test to support the claim.
* Results are shown in a mixed way: some on a table, some on the text. This makes it difficult to understand what is the overall performance.
* No information to relate the aaFD to time or fraction of the cardiac cycle makes interpretation of the results difficult

**Deanonymize Review:**

yes

**Detailed Comments:**

1. The obtained accuracy is given in frames, however there is no information of the frame rate or the heart rate, which makes it difficult to interpret this error. It would be interesting to put the detection error as a fraction of the heart cycle length, or as a fraction of the ED to ES distance. This might also explain large standard deviations, if there is a difference in frame rate between the different video sources, or differences in heart rate between patients.

2. Why are not all results shown in table 1 (adding multibeat and EchoNet datasets) and then use the space in the text to discuss what these results mean?

3. The amount of data used (quite large) allows for a statistical significance analysis, which I think is required particularly given the  relatively large standard deviations.

Minor comments:
3. Figure 1: The regression unit is not indicated in the figure, please add.





**Justification Of The Rating:**

I think the paper is very interesting and the method is clearly described, with code and data made available which is definitely a plus. There are some minor flaws, particularly interpretability of the results and clarity of how the results are presented, but I believe this can be easily corrected.

**Paper Type:**

both

**Special Issue:**

no

---

### Meta-Review · Program_Chairs · 2021-05-09

**Recommendation:** Accept (Poster)
**Confidence:** 4

**Metareview:**

Reviewers are unanimous in their recommendation to accept this paper.

---

### Decision · Program_Chairs · 2021-05-11

Accept (Poster)